# Non-Invasive Assessment of Back Surface Topography: Technologies, Techniques and Clinical Utility

**DOI:** 10.3390/s23208485

**Published:** 2023-10-16

**Authors:** Bhavna Mehta, Nachiappan Chockalingam, Thomas Shannon, Nikola Jevtic, Filip Lazic, Vinay Jasani, Nicola Eddison, Aoife Healy, Robert Needham

**Affiliations:** 1Centre for Biomechanics and Rehabilitation Technologies, Staffordshire University, Stoke on Trent ST4 2DF, UK; bhavna.mehta@alderhey.nhs.uk (B.M.); tom.shannon@staffs.ac.uk (T.S.); nicola.eddison@staffs.ac.uk (N.E.); a.healy@staffs.ac.uk (A.H.); r.needham@staffs.ac.uk (R.N.); 2ScolioCentar, Novisad, 403916 Novi Sad, Serbia; njevticns@gmail.com (N.J.); lazicf@gmail.com (F.L.); 3Centre for Biomechanics, University Hospitals of North Midlands NHS Trust, Stoke on Trent ST4 6QG, UK; vinay.jasani@uhnm.nhs.uk; 4Royal Wolverhampton NHS Trust, Wolverhampton WV10 0QP, UK

**Keywords:** adolescent idiopathic scoliosis, bracing, surface topography, orthosis, spine, rasterstereography, moire technique, motion capture

## Abstract

(1) Background: Frequent exposure to ionising radiation is often used to determine the diagnosis of adolescent idiopathic scoliosis (AIS), a lateral curvature of the spine in those aged between 10 and 18 years, and a treatment plan according to Cobb angle. This narrative review outlines the clinical utility of surface topography (ST), a radiation-free imaging modality. (2) Methods: Publicly available databases were searched to yield literature related to ST. Identified articles were classified based on the equipment used and in order of how it was developed, i.e., historical, recent developments, and state-of-the-art developments. (3) Conclusions: ST is a reliable cost-effective non-invasive technique that provides an alternative to radiation-based imaging to aid with the diagnosis and potential screening of AIS. Several scanning methods are available, which allows ST to be used in several clinical environments. Limitations of inter-reliability and differences of apparatus resulting in variations of data have been noted through this narrative review.

## 1. Introduction

Scoliosis is a deformity of the spine characterised as a three-dimensional (3D) rotation and lateral deviation of the vertebrae most prominent at the thoracic vertebrae (T1–T12) and lumbar vertebrae (L1–L5). Scoliosis can be characterised by the presentation of the spine in a ‘C’ or ‘S’ shape [1,2]. As a result, people diagnosed with scoliosis may experience cosmetic deformities in the form of, but not limited to, distorted rib structure, asymmetry of the shoulders/chest/hips, visible curvature of the spine in the coronal plane, prominent scapula or rib hump, and leaning of the body to one side [3,4].

The clinical diagnostic criterion and monitoring for scoliosis is a result of a Cobb angle equal to or greater than 10° in standing coronal radiographs [3,5,6]. This is considered the gold standard for scoliosis diagnoses [7]. However, this method provides only a two-dimensional (2D) classification of scoliosis and does not take into consideration the severity in the sagittal plane. From the Cobb angle, the King classification system deduces the type of curvature into five types in accordance with the shape and direction of tilt [8]. Due to the diagnoses and monitoring of AIS requiring X-rays or magnetic resonance imaging (MRI), patients are subjected to frequent exposure to ionising radiation, thus increasing the risk of carcinogenesis and other radiation-related health issues, such as nausea [9,10,11].

Adolescent idiopathic scoliosis (AIS) (Figure 1) is the most common scoliosis diagnosis: widely reported to affect up to 3% of the population aged 10–18 years [3,12,13,14]. The ratio between genders—female and male—and diagnoses at a curvature of between 11° and 20° have been reported to be between 1.4:1, with the prevalence being 1.5–3%. Although the prevalence decreases between 21° and 40° and 0.2 and 0.5%, the ratio between females and males increases to 2.8–5.4:1. Furthermore, this ratio increases at angles greater than 40° to 7.2:1, although the prevalence decreases further to 0.04–0.3% [15]. Patients diagnosed with moderate AIS (25°–40°) receive non-surgical treatment, whereas those with severe AIS (>40°) receive surgical treatment if non-surgical treatments do not rectify the deformity [16].

A considerable effort has been made to develop the effective and reliable use of radiation-free, non-invasive and low-cost measurement methods, such as surface topography (ST), in the assessment and monitoring of AIS. It is of importance to develop a method for the diagnosis and monitoring of scoliosis to reduce the ionising radiation that subjects are exposed to, thus reducing the risk of subjects developing cancerous diseases. A trained clinician does not have to use any additional apparatus or equipment, thus making ST a suitable choice within several clinical environments. These methods generate a 3D/4D structure of patients’ backs using various models and protocols, which can quantify the cosmetic deformity of AIS and do not subject patients to ionising radiation [17,18].

ST has been noted to be reliable in detecting the progression of the scoliosis curve well in a coronal, transverse and sagittal view for decompensation, trunk/imbalance angle and lordosis, and the lateral deviation of curve and rotation [19,20], and not be clinically very different in resultant outcomes when compared to radiographic measurements [21]. However, the costs, intra- and inter-rater reliability and correlation with a radiographically acquired Cobb angle have made such apparatuses’ clinical application equivocal. As various models of ST are available, with research having assessed their correlation, reliability and reproducibility, this review aims to assess the historical use of ST, its current trends and the viability of previous and current models being used. The authors will make a recommendation as to which method of ST is appropriate for clinical assessment and monitoring of scoliosis.

## 2. Methods

The authors conducted a narrative overview as it was deemed the most appropriate method for providing a comprehensive description of the diverse spectrum of available ST technologies and techniques. Given the scarcity of large-scale studies with extensive sample sizes conducive to systematic reviews or meta-analyses, the primary objective of this paper is to offer an up-to-date summary of the current state of the field. Unlike alternative review methodologies, such as systematic reviews, which focus on specific inquiries or methodologies, our narrative review approach facilitated the derivation of recommendations based on suitability for clinical applications. This approach is expected to enhance the informed utilisation of ST, subsequently paving the way for more structured and purposeful investigations in this domain. Initial searches were conducted electronically on the Ovid and PubMed databases using the keywords ‘scoliosis’ OR ‘adolescent idiopathic scoliosis’ AND ‘surface topography’ OR ‘dynamic surface topography’ AND ‘radiation-free’ OR ‘non-radiographic’. Additionally, the keywords ‘scoliosis’ OR ‘adolescent idiopathic scoliosis’ and ‘Moiré’ OR ‘rasterstereography’ were searched. Using these terms allowed for a historical overview to be provided and articles to be identified that implemented novel methods of ST, which did not expose subjects to ionising radiation. Papers that focused on analogue measurements, like scoliometers or radiographs, as well as other electronic instrumentation, such as a goniometer, were not considered. The included papers specifically utilised optical methods to evaluate surface topography. In addition, the review also includes papers focusing on motion capture and methods using ultrasound for the assessment of spinal curvature.

## 3. Results

Previous research has examined the correlation and reliability of different apparatuses and methods, such as the Moiré technique, rasterstereography, motion capture and ultrasonography, in relation to the “gold standard” radiographs. However, the existing literature primarily focuses on describing the equipment used in these studies, primarily due to the novelty of the equipment and limited accessibility elsewhere. Notably, the studies identified did not compare the different surface topography (ST) methods to one another. This lack of comparison could be attributed to factors such as clinical acceptance, equipment availability and the continued reliance on radiographs for diagnosing and monitoring scoliosis.

### 3.1. Historical Overview

With the advent of various apparatuses, there has been a variety of methods developed to subjectively quantify the degree of deformity [22,23], as well as algorithms that objectively define the degree of deformity when assessed using surface topography [24,25,26]. The Walter Reed Assessment Scale provided patients with a visual method to subjectively quantify their cosmetic deformity, which required patients to score various domains from 1 to 5, with greater deformity reflected with a higher score. Despite being a subjective measure, there was a significant correlation between the scores obtained from patients and curve magnitude, thus providing a method with which to obtain a quantification for cosmetic deformity [27].

Radiography measures the severity of the curvature of the spine, but it does not necessarily measure the cosmetic deformity subjects experience as a result of possessing AIS [23,28]. During adolescence, there may be a greater emphasis on the subject’s part to rectify the cosmetic deformity more so than the spinal curvature [2]. This has been demonstrated by research assessing the mental health behaviours of AIS subjects, particularly female adolescents [22,23,28,29]. It was identified that cognitive distress, body image and anxiety were negatively impacted upon the diagnosis of scoliosis, but these indicators were markedly improved following intervention via bracing. This further highlights the necessity of developing non-invasive techniques that quantify back shape deformity and asymmetry based on reducing exposure to ionising radiation and increasing the mental health of patients. Such techniques and apparatus have been developed for and in schools, and sometimes implemented clinically, ranging from a simple inclinometer to the more sophisticated rasterstereography.

Historically, surface topography stems from the development of the Moiré technique, which has been implemented clinically since 1970 in the screening of scoliosis [30,31]. This technique projects light through two overlapping patterned grids onto the back of a subject standing in an anatomical position, with these projections captured photographically [32]. The distance from the light source and camera in relation to the subject and differences in the contour lines of a participant’s back can then be analysed [30,31]. It has proven beneficial due to being non-invasive and radiation-free [31]. Whilst the papers argue that Moiré is faster and more economical [31], there are conflicting statements regarding the accuracy of the results of the Moiré technique due to the lack of well-defined methodological procedures in the literature [30,31,33]. Therefore, the papers call for Moiré techniques to complement radiography rather than provide an alternative. However, it can be noted that it reduces the dosage of radiation patients are exposed to.

Due to the increased availability and affordability of computing systems and digital video recording technologies, there were technological developments in the apparatus, allowing for the quantifiable measurement of surfaces during the 1980s, which culminated in the development of rasterstereography in 1990s based on the principle of triangulation [34]. Rasterstereography is a radiation-free, non-invasive stereophotogrammetric surface measurement, whereby a slide projector projects white gridlines or horizontal lines onto the subject’s back, which are then photographed within seconds. Due to the 3D shape of the back, the camera is able to capture deformations of the gridlines [35]. Rasterstereography is able to generate a 3D reconstruction of the subject’s back through the identification of anatomical landmarks and deformation of the gridlines without exposure to ionizing radiation [2,24,36]. However, with an increasing amount of technology becoming available there is uncertainty surrounding the applicability of this technique for clinical use.

Rasterstereography has spawned a plethora of apparatus that employ various methods and algorithms. These apparatuses include the following: The Integrated Shape Imaging System (ISIS) (Oxford Metrics Ltd., Oxford, UK); the Quantec© system (Quantec Image Processing Ltd., Liverpool, UK); and ISIS2 (Oxford Metrics Ltd., Oxford, UK). ISIS was able to determine if significant curve progression had occurred, which informed whether subjects required surgical or non-surgical treatment. Additionally, the estimated Cobb angle, as predicted by ISIS over a 2-year period, had a high correlation coefficient (r = 0.91, *p* < 0.0001) and was suggested to have an accurate estimation of the rate of progress of scoliosis [37]. ISIS2 was developed later on and adopted digital photography as opposed to a scanned beam of light used by ISIS. This reduced acquisition time from 0.5 s to 100 ms, which reduced breathing and sway influence. Additionally, the ISIS2 analysis time was reduced from 10 min to around 40 s compared to ISIS, however, inter-rater variability ranged between 5° and 10° [38]. Recent longitudinal studies employing ISIS2 have detected divergent patterns in the progression of kyphosis and lordosis among adolescents without [39] and with [40] scoliosis. Also, the Quantec© system was able to complete acquisitions in a fraction of a second, which reduced the influence of breathing and postural sway. However, due to these apparatuses conducting single sample acquisitions, they were prone to postural, stance and breathing errors between acquisitions. These apparatuses have provided methods in which scoliosis can be monitored, although they have not been adopted clinically as they have not been found to be necessarily low-cost or generate reproducible results due to their lack of accountability for sway, breathing and stance. Additionally, these apparatuses provide feedback on the basis a subject is static and do not measure the implications during dynamic tasks, such as walking.

### 3.2. Recent Developments

The Formetric 3D system (Diers^®^ International GmbH, Schlangenbad, Germany) adopted rasterstereography and is mathematically similar to ISIS in how it estimated the distortion or lateral deviation of subjects’ backs. However, errors would occur due to postural sway and breathing during acquisition. Although the Formetric 3D system was recommended for its possible clinical implementation despite subject posture having influenced its reliability [41], before it was widely adopted clinically, the Formetric 4D system (Diers^®^ International GmbH, Schlangenbad, Germany) was developed, also implementing rasterstereography, which acquires 12 digital images during a 6 s static scan [42,43,44]. The software used in the Formetric 4D system approximated bony landmarks and asymmetries in comparison to previously stored radiographic scans, and the scans were averaged to counter postural or breathing sway during acquisitions [43]. However, it was stated the obese participants required palpation and marker placement to identify bony landmarks prior to scanning, which could affect results based on inter-rater reliability. Additionally, similar to the previously mentioned rasterstereography apparatus, the Formetric 4D system does not acquire scans during dynamic tasks. However, this method has been found to be comparable to radiography in test–retest reliability and correlates strongly with the Cobb angle but is unable to quantify curve magnitude [43]. It was suggested that this method of rasterstereography would be more suited to the monitoring of scoliosis as opposed to the diagnosis, but the cost of implementation clinically may prove negligible just for monitoring purposes. An alternative commercially available system, known as BIOMOD™- 3S (AXS MEDICAL, Merignac, France), utilizes optical techniques to measure the 3D surface topographic parameters. This system has demonstrated efficacy in evaluating scoliosis, and research indicates that its implementation can lead to a 30% reduction in the number of required X-rays [45].

The available research on these systems has provided some insights, yet there remains a notable gap in understanding their cost-effectiveness. Studies suggest that while these techniques show comparable test–retest reproducibility to standard radiographs, it tends to underestimate various spinal parameters when compared to conventional radiography [41,43]. However, these techniques do play a role in reducing the frequency of radiographic examinations needed to monitor AIS curve progression and there is a growing interest in optimising their timing and efficacy. They prove valuable in diverse scenarios, encompassing initial assessments when the necessity for radiographs is uncertain; sequential examinations to monitor deformity progression; and the accurate quantification of the magnitude, three-dimensional shape and rate of advancement of spinal deformity [30]. Standardising the application of surface topography, particularly through the use of surface markers, may enhance the accuracy of mathematical models, especially in the presence of additional factors, such as skin and subcutaneous lesions or scars.

### 3.3. State of the Art Developments

An apparatus that has been predominantly implemented in gait analysis and rehabilitation is a multi-camera optoelectronic motion capture system. Using this technology, previous research has evaluated the effect on the gait cycle for those with scoliosis [46,47,48,49]. Despite the ability of these systems, it is a high-cost apparatus and may not be able to be adopted clinically to identify asymmetries in scoliotic subjects during dynamic tasks. In addition to appropriate techniques, the results obtained can be correlated with the Cobb angle. Schmid et al. [46] highlighted that the marker-based system may underestimate curvature angles compared to radiographically acquired Cobb angle measurements, and that marker placement could affect the test–retest ability in the continued assessment of scoliotic patients. One should also note that the planar placement of cameras could cause markers to become obscured, thus further affecting angle measurements. Additionally, the limitations of motion capture are the space and clinical capacity to enable this within an outpatient clinical environment. Recently optical capture technology, such as the Kinect™ V1 and V2 (Microsoft Corporation, Washington state, USA), which was originally intended for use with the Xbox™ gaming console, have been utilised [50,51,52]. This apparatus uses time-of-flight depth sensor technology and can track 20 markerless bony landmarks at 30 Hz using an estimation algorithm [51,53] to measure form and function. The Kinect™ apparatus has been evaluated in its accuracy in assessing clinical torso and spinal deformity [50] (Figure 2), and it was found that the Kinect™, a commercially available and low-cost apparatus, produced an acceptable agreement of standard deviation in linear surface deviations compared to a higher-cost and less portable apparatus at +1.58 ± 1.50 mm and −0.58 ± 0.58 mm, respectively. Additionally, it was found to be within a clinical measurement variability, thus deduced to be suitable for clinical use. However, it is suggested that due to the utilisation of infrared, the Kinect™ would be susceptible to degradation. Müller et al. [54] assessed the Kinect™ system in gait analysis and found that a setup with several Kinect™ sensors placed in various planes around a subject during walking tasks achieved excellent agreement of spatiotemporal gait parameters when compared to optoelectronic systems. Additionally, it has been identified that the use of a novel marker-based Kinect™ setup using a laterally placed sensor achieved a good agreement with optoelectronic systems during various walking speeds during gait analysis. Recent studies have employed this approach to investigate the association between surface topography (ST) and curve type [55], as well as to assess the cosmetic changes resulting from exercise interventions [11]. Another recent study assessed the reproducibility of sagittal spine curvature measurement using the Microsoft Kinect sensor [56]. The results demonstrated high reliability both within and between raters, with the second image capture showing the most consistency. With a sample size of 37, the authors suggest that the Kinect™ offers a reliable and efficient method for clinical and research applications in assessing spinal curvature [56].

Over the years, optical or light-based scanners have been widely used for various types of 3D non-contact scanning, such as in engineering and architecture, computer animation and reconstructions. Within medicine, there is significant interest in utilising non-contact scanners to capture surface anatomy for purposes, like orthotic devices and plastic surgery reconstructions. Recently, there has been an increasing interest in using 3D scanning to evaluate spinal shape for the biomechanical and ergonomic assessments of spinal posture during various activities. Surface scanning is currently employed in clinical settings to assess torso shape in patients with deformities or to create customized spinal braces. In recent studies, researchers have examined the changes in vertebral positioning using non-contact scanning and compared them to MRI results. The findings suggest that surface markers could potentially serve as a valuable method for measuring internal changes in sagittal curvature or skeletal transformations [50,57]. Other systems, such as the 3dMDbody system (3dMD, Atlanta, GA, USA), also use similar technologies. Recent studies utilizing this system have shown a noteworthy distinction in axial motion between individuals with and without scoliosis. The research findings further indicate significant differences, both in cases of severe scoliosis compared to the control group and in cases of thoracic scoliosis compared to controls [58]. Another study conducted using the same system reveals that patients with severe structural deformity and restricted range of motion do not exhibit a decrease in their physical activity level, as indicated by validated patient activity questionnaires [59]. Additional studies have demonstrated a strong correlation between 3D surface topographic measurements for idiopathic scoliosis and patient self-image questionnaires [60].

In addition to the surface-based systems, there are other systems that utilise ultrasound to gather information. These ultrasound-based systems do not involve the use of ionising radiation and have the capability to provide insights into the underlying skeletal structures. One such system called Scolioscan (Telefield Medical Imaging Ltd., Hong Kong) [61] has been noted to provide accurate Cobb angles in comparison to radiographic imaging (thoracic: R^2^  =  0.959, lumbar: R^2^  =  0.936, *p*  <  0.001) [62]. The Scolioscan apparatus measures patients in an upright position similar to rasterstereography apparatuses, while rigid chest and hip boards and a green-eye spot for subjects are used to maintain a neutral head position. Participants’ backs are scanned freehand using an ultrasound probe dictated by parameters based on subject information, which generates a 3D model of the spine via B-mode image data and corresponding position and orientation data. This radiation-free method has proved ‘very good’ intra and inter-rater reliability with ICC values of 0.97 ± 0.2 and 0.90 ± 0.02, respectively. Additionally, there were moderately linear correlations (R^2^ > 0.72) between the obtained Scolioscan angles and Cobb angles, with the Scolioscan determined to ‘slightly’ underestimate spinal deformity in the thoracic and lumbar regions. It has been noted that rasterstereography is a reliable method for mild AIS [63].

In comparison to the previously mentioned rasterstereography apparatuses, the Scolioscan could be considered as having a more time-consuming acquisition due to rasterstereography taking a snapshot of subjects in milliseconds. The intra-reliability of Scolioscan with the X-ray analysis of the Cobb angle has been investigated for the thoracic and lumbar regions for linear correlation and tested via the Bland–Altman method, with a standard deviation of around 12 degrees [61]. However, limitations of Scolioscan include the fact that it does not measure patients undertaking dynamic tasks. Furthermore, the effectiveness of Scolioscan on clinically obese subjects may be altered due to layers of soft tissue and a reduced ultrasound frequency, possibly resulting in reduced image resolution; thus, this may affect angle measurement accuracy. Therefore, this ultrasound technique may not be affected by sway, breathing or stance due to the acquisition compared to rasterstereography.

## 4. Discussion

It has been identified that radiography remains the ‘gold standard’ in the diagnosis and measurement of the severity of scoliosis, monitoring the progression of the disease and informing treatment implementation and effectiveness. However, radiography is accompanied with several limitations: exposure of subjects to ionising radiation, thus increasing the possibility of developing cancerous diseases, potentially impacting their quality of life; radiographs only provide a 2D representation of a 3D deformity, which does not quantify the severity of deformity and are unable to assess the effects of scoliosis on dynamic tasks; and inter-observer and intra-observer errors are within reach of what is considered clinically unacceptable. As a result of these limitations, there has been a considerable effort in the research to identify a clinically significant correlation between scoliosis and ST, but until recently, such research found ambiguous correlations and that apparatus was prone to error and were not sufficiently robust in identifying scoliotic deformity and therefore were not appropriate for clinical implementation.

The Moiré technique provided a seminal method as an alternative to radiography [30,31]. However, it can be reported to be unreliable as it is difficult to undertake the diagnosis of rotation [64]. The Moiré technique relies on the interference patterns created when a regular grid or pattern is overlaid on a surface with curvature. By observing how the grid deforms, it is possible to infer information about the shape of the underlying surface. However, Moiré patterns are primarily effective for quantifying deformations in one plane. When it comes to rotational deformations, the technique lacks the necessary precision and sensitivity to accurately capture and measure these types of changes. This technique was found to therefore be a complementary method alongside radiography rather than a replacement due to the breathing and sway artefacts during acquisition, which would result in errors and the cost of implementation. It did however provide a radiation-free, non-invasive method in the assessment of scoliosis. Following research into the effectiveness of the Moiré technique, radiography continued to be applied clinically, and research into alternatives to radiography was not continued until several years after the development of the Moiré technique, which informed the apparatuses and methods used in more recent developments.

Rasterstereography was developed with the intention of subjectively quantifying cosmetic deformities, thus enabling the ability to address subject cosmetic concerns whilst also being able to identify any physical impairments and the effects on quality of life. Earlier iterations of rasterstereography, such as the ISIS, were able to identify if significant curve progression had occurred over a period of time and could accurately estimate the rate of progression. However, such conclusions were made when measured amongst only AIS subjects and not older subjects. Although the standout limitation was the acquisition time and how breathing and postural sway impacted upon the results, which was addressed in later iterations of rasterstereography, including the ISIS2, Quantec©, Biomod 3S, and Formetric 3D and 4D by reducing the acquisition time, which reduced the influence of artefacts, this would not necessarily affect the impact of postural sway within-acquisitions. However, the study by Degenhardt et al. [44] compared the results of human subjects to a mannequin in the appraisal of the Formetric 4D, which removed the implications of postural and breathing sway on results, thus potentially affecting the outcome.

Despite rasterstereographic apparatuses being found to identify cosmetic deformity and acceptably estimate curve progression in comparison to the Cobb angle, they were not adopted clinically, which may be due to the angles not being an exact match in relation to radiograph acquisitions and the inter-rater reliability errors, as a result, may not have been cost-effective to implement [65]. Additionally, the studies that assessed these apparatuses stated that reliability and estimation errors may be exacerbated in obese subjects; thus, any reliability and correlations found are not necessarily applicable to the overweight population. Also, Degenhardt et al. excluded subjects with a body mass index (BMI) either above 35 or below 20 and those with a back tattoo, which could suggest those considered slightly either underweight or overweight, or have a tattoo could affect the outcome. As a result, clinics continue to utilise qualitative assessment methods in the identification of cosmetic deformities, such as the Adams forward bend test or a handheld inclinometer, which use subjective analysis and are prone to errors due to position, posture, stance, sway and breathing artefact.

Motion capture technology provides a method to assess scoliosis either with markers (optoelectronic systems such as Vicon) or without markers (Kinect™) in comparison to radiographs. However, the results of the optoelectronic apparatus appear to be negatively influenced slightly by the marker system, whereas the Kinect™ apparatus benefitted from the implementation of marker placement. Another motion capture method reported to be complementary to diagnostic imaging is the 3dMD scanner which uses cameras to capture video technology at 10 frames per second at a speed of 1.7 mm/s [66]. Motion capture technology (Vicon and Kinect™) as well as the 3dMD scanner and Scolioscan has yet to be used to assess the implications of scoliosis on dynamic tasks. Also, these apparatuses are able to acquire surface and bony landmark data as well as produce averaged measurements, which account for artefacts that would otherwise affect the results of single acquisitions and within-acquisition of Moiré and rasterstereography apparatuses. Additionally, motion capture could provide a clinically significant method to assess and monitor scoliosis due to the availability of apparatuses in the likes of orthopaedic clinics, which conduct gait analysis and could be adjusted to assess for trunk deformity.

Although previous studies have established that there is no definitive correlation for all curve types between the progression of scoliosis and changes in body asymmetry, for many patients, the motivation in seeking treatment is the improvement of their appearance rather than to correct an underlying skeletal deformity [55]. Cosmetic concerns, quality of life, and the psychosocial impacts of adolescent idiopathic scoliosis remain important factors in any clinical decision-making process [2]. In conjunction with conventional radiologically derived measures, the addition of validated body asymmetry metrics should be included when designing patient-specific treatment plans and assessing outcomes. Consolidating radiological diagnoses (Cobb angle, curve type) and clinical assessment data from groups of patients has additional value in the longitudinal assessment of the relative efficacy of differing clinical interventions within physical medicine.

## 5. Conclusions

This paper provides a historical overview of ST and its use within the diagnostic identification of AIS. There are limitations of inter-reliability and differences of apparatus resulting in variations of data produced. However, ST is a notable and reliable non-invasive technique able to provide a radiation-free alternative that any clinician can offer to evaluate trunk cosmesis. ST is also particularly highlighted in use as part of school screening, whereby it can be used in various clinical environments to provide diagnostic criteria for scoliosis. There are limitations of motion-capture-related ST, as dynamic movements cannot be accurately measured. Other techniques, however, such as Moiré and rasterstereography require landmarks to enable the production of data and have been noted as being unreliable for those that are clinically obese and can be noted to be difficult to undertake parameter setting.

## Figures and Tables

**Figure 1 sensors-23-08485-f001:**
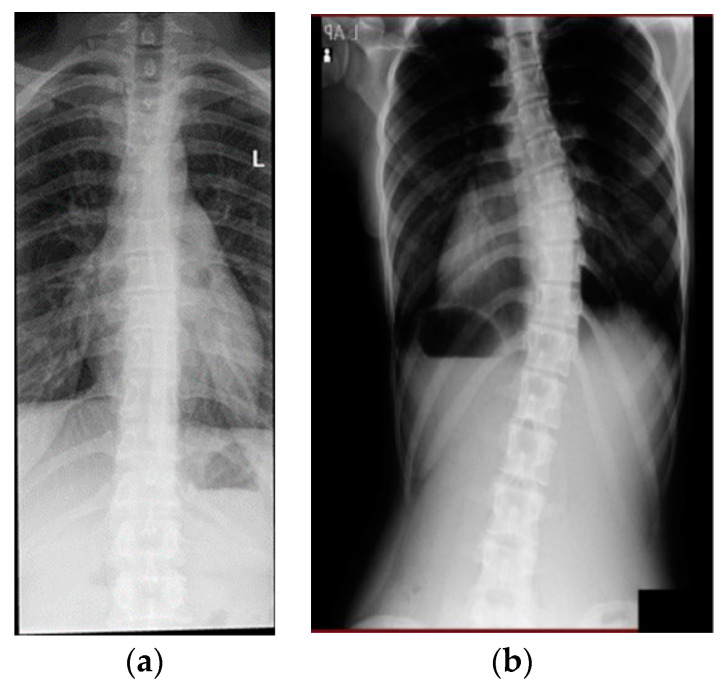
(**a**) Normal spine (case courtesy of Craig Hacking, Radiopaedia.org, rID: 40136, accessed on 9 October 2023). (**b**) Adolescent idiopathic scoliosis.

**Figure 2 sensors-23-08485-f002:**
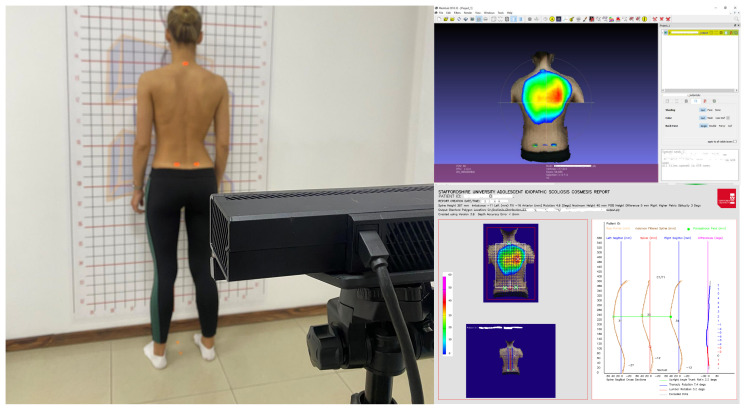
Kinect setup in a clinic and its outputs.

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
