# Peer review of "Non-Invasive Assessment of Back Surface Topography: Technologies, Techniques and Clinical Utility"

_sensors, 2023, doi:10.3390/s23208485_

Round 1

Reviewer 1 Report

Thanks for the possibility to review your work. I have some revisions I would kindly ask authors to address.

1 It is recommended to highlight the key points and appropriately delete the content of keywords.

It is recommended to add a normal spine to Figure 1 for comparison in Section 1.

It is recommended to provide a more direct proof of choosing a narrative review as the best option in Section 2.

4 It is recommended to provide a specific explanation in the third paragraph of Section 3.1 regarding the conflict between the cost-effectiveness and accuracy of Moiré technology results.

It is recommended to provide more detailed information in the fourth paragraph of Section 3.1 when describing the equipment, such as the working principle, data collection methods, and data analysis methods of each device.

6 It is recommended to supplement information on the working principles, limitations, cost-benefit analysis, and feasibility in clinical applications for the three systems in Section 3.2.

It is recommended to provide more practical cases to prove its feasibility in practice for Kinect Technology in the first paragraph of Section 3.3.

It is recommended to explain how Moiré technology works, so that readers can better understand why it is difficult to diagnose rotation in the second paragraph of Section 4.

9 It is recommended to provide more specific research results and data support on the correlation between appearance improvement and body asymmetry in patients with scoliosis in the last paragraph of Section 4.

10 It is recommended to supplement the usage trend and areas for improvement of ST in Section.

The overall description of the article is smooth, with only a few sentences and grammar that need to be optimized

Author Response

Thanks for the possibility to review your work. I have some revisions I would kindly ask authors to address.

1 It is recommended to highlight the key points and appropriately delete the content of keywords.

We don’t fully understand this comment. We have followed the instructions provided by the journal and we are happy to provide key points.

Key points

  • This paper presents a historical overview of Surface Topography (ST) and its application in the management of Adolescent Idiopathic Scoliosis (AIS).
  • Techniques like Moiré and rasterstereography require landmarks for data production and may prove unreliable for clinically obese individuals and involve difficulties in parameter setting.
  • Latest ST technologies and techniques involving optical methods emerges as a noteworthy and reliable non-invasive diagnostic technique, providing a radiation-free alternative to X-ray imaging.
  • Clinicians can readily incorporate ST in school screenings and various clinical environments to establish diagnostic criteria for Scoliosis.
  • Challenges exist in terms of inter-reliability and variations in data due to differences in the apparatus used for ST.

2 It is recommended to add a normal spine to Figure 1 for comparison in Section 1.

Thank you for your suggestion. We have now included a picture of a normal spine.

3 It is recommended to provide a more direct proof of choosing a narrative review as the best option in Section 2.

Thanks for this suggestion. We have modified the opening sentence in section 2, to reiterate our point.

4 It is recommended to provide a specific explanation in the third paragraph of Section 3.1 regarding the conflict between the cost-effectiveness and accuracy of Moiré technology results.

 The sentence is now modified and expanded with appropriate references.

5 It is recommended to provide more detailed information in the fourth paragraph of Section 3.1 when describing the equipment, such as the working principle, data collection methods, and data analysis methods of each device.

Thanks very much for your comment. However, we feel that it is beyond the scope of this manuscript to provide a detailed explanation and working principles of all of the described methods. We have provided appropriate references against each technique for the reader to explore further.

6 It is recommended to supplement information on the working principles, limitations, cost-benefit analysis, and feasibility in clinical applications for the three systems in Section 3.2.

Whilst the description of working principles might be beyond the scope of the manuscript, we have now added a paragraph to expand this section.

7 It is recommended to provide more practical cases to prove its feasibility in practice for Kinect™ Technology in the first paragraph of Section 3.3.

We have now added another reference to indicate the reliability and validity in measuring spinal curvature.

8 It is recommended to explain how Moiré technology works, so that readers can better understand why it is difficult to diagnose rotation in the second paragraph of Section 4.

 We included text to expand this section.

9 It is recommended to provide more specific research results and data support on the correlation between appearance improvement and body asymmetry in patients with scoliosis in the last paragraph of Section 4.

 We have now included references in this section.

10 It is recommended to supplement the usage trend and areas for improvement of ST in Section.

We are not sure what this comment means but we hope the other changes to the manuscript would have addressed this comment.

Reviewer 2 Report

General impression

In this narrative review, the authors reported the historical use of surface topography (ST), its current trend, and viability of previous and current models being used.  And they conducted ST is a reliable cost-effective non-invasive technique that provides a suitable alternative to radiation-based imaging to aid with diagnosis and progression of adolescent idiopathic scoliosis.

I evaluate this paper well summarized the ability and potential of some apparatuses related ST to actual clinical field, however, I think they must be overestimation.  Currently, treatment strategy of scoliosis is mainly decided on the basis of Cobb angle.  Radiographical imaging must be essential to measure accurate Cobb angle.  I guess these apparatuses should be limited the use for screening to detect scoliosis or evaluation of trunk cosmesis.

However, the methodology of this study was precisely explained.  And I could not find either error in writings or mistakes in the text.  Therefore, I would judge this manuscript can be accepted and published by sensor journal after revision of a couple requests to be revised as stated below.

*I ask the authors that correction parts will be shown in red color in the revised manuscript.

1. I hope the conclusion will be altered by reference to my concern as I stated in the general impression if the authors can agree.

2. I recommend some pictures of apparatuses related ST (ISIS, Quantec system, Scolioscan) just like as “Kinect”.  I believe they must facilitate the understanding and the imaging of readers.

Author Response

General impression

In this narrative review, the authors reported the historical use of surface topography (ST), its current trend, and viability of previous and current models being used.  And they conducted ST is a reliable cost-effective non-invasive technique that provides a suitable alternative to radiation-based imaging to aid with diagnosis and progression of adolescent idiopathic scoliosis.

I evaluate this paper well summarized the ability and potential of some apparatuses related ST to actual clinical field, however, I think they must be overestimation.  Currently, treatment strategy of scoliosis is mainly decided on the basis of Cobb angle.  Radiographical imaging must be essential to measure accurate Cobb angle.  I guess these apparatuses should be limited the use for screening to detect scoliosis or evaluation of trunk cosmesis.

However, the methodology of this study was precisely explained.  And I could not find either error in writings or mistakes in the text.  Therefore, I would judge this manuscript can be accepted and published by sensor journal after revision of a couple requests to be revised as stated below.

 Thank you for these positive comments.

*I ask the authors that correction parts will be shown in red color in the revised manuscript.

  1. I hope the conclusion will be altered by reference to my concern as I stated in the general impression if the authors can agree.

Thanks for your comment. The changes to the manuscript based on the comments from Reviewer 1 addresses the issue raised. We have also modified the sentences within the conclusion sections both in the abstract and in the main manuscript to tone down the claims.

  1. I recommend some pictures of apparatuses related ST (ISIS, Quantec system, Scolioscan) just like as “Kinect”.  I believe they must facilitate the understanding and the imaging of readers.

Thanks for this comment. We tried to get these pictures but due to copyright issues we haven’t been able to reproduce them. However, the references used can provide enough information to the reader.

Round 2

Reviewer 1 Report

In the manuscript, the authors have made effective modifications to my recommendations. In my opinion, this manuscript can be accepted in present form.